# Evaluation of Myo-Intimal Media Thickness and Atheromatous Plaques in People Living with HIV from the Archiprevaleat Cohort vs. HIV-Negative Subjects

**DOI:** 10.3390/biomedicines12040773

**Published:** 2024-04-01

**Authors:** Salvatore Martini, Elena Delfina Ricci, Addolorata Masiello, Sergio Zacà, Benedetto Maurizio Celesia, Sergio Ferrara, Giovanni Di Filippo, Alessandra Tartaglia, Rosa Basile, Domenico Angiletta, Paolo Maggi

**Affiliations:** 1Department of Mental Health and Public Medicine, Section of Infectious Diseases, University of Campania, Luigi Vanvitelli, 80131 Naples, Italy; paolo.maggi@unicampania.it; 2Fondazione ASIA Onlus, 20090 Milan, Italy; ed.ricci@libero.it; 3AORN Sant’Anna e San Sebastiano of Caserta, 81100 Caserta, Italy; dora.80@live.it; 4Department of Emergency and Organ Transplantation, University of Bari School of Medicine, 70121 Bari, Italy; sergiozac89@gmail.com (S.Z.); domenico.angiletta@uniba.it (D.A.); 5Unit of Infectious Diseases, Department of Clinical and Experimental Medicine, ARNAS Garibaldi Hospital, University of Catania, 95123 Catania, Italy; bmcelesia@gmail.com; 6Department of Medical and Surgical Sciences, Section of Infectious Diseases, University of Studies of Foggia, 71122 Foggia, Italy; sferrara@ospedaliriunitifoggia.it; 7Department of Medicine and Surgery, Section of Infectious Diseases, University Federico II of Naples, 80138 Napoli, Italy; giodifi@alice.it; 8Azienda Ospedaliera di Foggia, 35128 Foggia, Italy; alessandratartaglia@yahoo.it; 9Section of Infectious Diseases, Grande Ospedale Metropolitano, Bianchi Melacrino Morelli, 89124 Reggio Calabria, Italy; rosa.basile@ospedalerc.it

**Keywords:** HIV, ART, Doppler, IMT, plaques

## Abstract

Background: Antiretroviral therapy has allowed a clear improvement in prognosis for HIV patients, but metabolic problems, such as dyslipidemia, remain. This can lead to the development of atheromatous plaques. Our study aims to evaluate whether HIV-positive (HIV+) patients show higher myo-intimal media thickness (IMT) and atheromatous plaques compared to HIV-negative (HIV−) patients. Methods: To evaluate the association between HIV infection in experienced patients and vascular pathology, we performed a cross-sectional study, observing 1006 patients, 380 HIV+ enrolled in the Archiprevaleat cohort, and 626 HIV− as a control group. All patients underwent a Doppler scan of the supra-aortic vessels. We compared the prevalence of IMT > 1.0 mm and plaques in the two groups. Results: Patients in the HIV+ group were younger than those in the HIV− group, with a lower prevalence of hypertension and diabetes and higher dyslipidemia. The prevalence of plaques in strata of age was higher in the HIV+ group than in the HIV− group and was associated with the length of ART exposure. Conclusions: Our cross-sectional, retrospective study shows that HIV+ experienced patients are at greater risk of IMT and atheromatous plaques compared to HIV−. The risk is associated with being HIV+ and with the length of ART exposure. This finding may be useful in preventing cardiovascular risk.

## 1. Introduction

Antiretroviral therapy (ART) has allowed HIV patients a clear improvement in prognosis and quality of life, but metabolic problems remain, characterized mainly by dyslipidemia, often associated with a progressive myo-intimal thickening of the supra-aortic vessels. This condition can induce the development of atheromatous plaques, causing increased cardiovascular risk (CVR). The progressive improvement in the efficacy of antiretroviral regimens has changed the target of treatment, which is not only linked to viro-immunological efficacy but also metabolic safety to guarantee the durability of therapy and a better long-term prognosis. There is much data in the literature on the impact of metabolic alterations and atheromatous plaques in patients with HIV. Maggi et al. in a review, described the role of CVR and dyslipidemia in this context, demonstrating that this aspect remains one of the major problems in the management of HIV patients [1]. In another review and meta-analysis, Msoka et al. showed that HIV infection and ART seem to influence the progression of carotid intima-media thickness (cIMT) and configure itself as a risk factor for cardiovascular events [2]. In a recent study, Ngamdu et al. showed an association between the Framingham Risk Score (FRS) and cIMT in patients with HIV [3]. They enrolled, in a cross-sectional study, 96 experienced patients on stable antiretroviral therapy without a history of cardiovascular disease (CVD) who underwent carotid ultrasound evaluation. In multivariable analysis, only FRS (*p* = 0.009) was independently associated with increased cIMT. Kreek et al. described a higher percentage of cIMT at younger ages in HIV-positive patients compared to the reference values for an HIV-negative cohort [4]. Of the HIV-specific variables, a relationship was observed only between cIMT and length of antiretroviral therapy and between cIMT and (inversely) current ART. Higher cIMT was found in HIV-positive patients compared to controls. Unlike HIV-specific variables, classical CVR factors were associated with higher cIMT and should, therefore, be the focus of preventive measures. In another review and meta-analysis of real-life data, Grand et al. showed that people living with HIV (PLWH) have a 2-fold greater risk of having a cardiovascular event compared to HIV-negative individuals [5]. Data obtained in this systematic review indicate that more than one in five individuals with HIV have moderate–high CVR. Of course, dyslipidemia and cIMT in HIV may also be related to aging as occurs in healthy populations, but another recent study in HIV-positive children highlights the role of infection and antiretroviral therapy. Mukhuty et al., in fact, studied cIMT and metabolic complications in HIV-positive children treated with ARV compared to a control group of HIV-negative children [6]. This cross-sectional study concluded that children with HIV on ART have significantly higher cIMT and increased metabolic abnormalities. This is important to clarify the direct role of HIV and ART in the genesis of metabolic disorders associated with cIMT. The enrolled population, indeed, was based on children who could not have chronic vessel damage from aging. Furthermore, the children, given their young age, may not have a long history of ARV treatment. Despite this, they developed more metabolic complications and cIMT than the control group of healthy children. The aim of the present study is to evaluate the prevalence of plaques in this cohort of PLWH, as compared to a group of HIV-negative subjects, investigating the potential role of HIV infection and ART treatment besides that of the classic CVR factors.

## 2. Materials and Methods

To evaluate the association between HIV infection in experienced patients and vascular pathology, we performed an observational, retrospective, cross-sectional study enrolling 1006 patients, 380 HIV-positive persons (HIV+) from the Archiprevaleat cohort and 626 HIV-negative (HIV−) as a control group (Figure 1). The Archiprevaleat cohort is described in a previous article [7]. Briefly, Archiprevaleat is a National Registry of color Doppler ultrasound, created to evaluate the characteristics of vascular lesions in PLWH on a large number of data. The project involves Italian centers where the examination is carried out by specially trained clinicians. The Registry is based on an online platform aimed at collecting data relating to cIMT and plaques in patients routinely subjected to the examination. The control group enrolled HIV-negative patients followed at the Vascular Surgery Unit of the University of Bari, Italy. These patients were routinely examined for various pathologies, performing Doppler scanning as screening. All patients in the two groups had a Doppler scan of the supra-aortic vessels between 2009 and 2023. We evaluated the following parameters:
IMT of common and internal carotid for both left and right sides, details have been previously published [7]. An IMT greater than 1.0 mm and/or atherosclerotic plaques (IMT > 1.2 mm) are considered pathologic findings [8]. Atherosclerotic plaques, if present, were described. All images were photographed and archived.Data regarding independent risk factors for cardiovascular disease (CVD), such as hypertension, dyslipidemia, and diabetes. Hypertension was defined as office systolic BP of ≥140 mmHg and/or a diastolic BP of ≥90 mmHg or receiving antihypertensive therapy at the time of the examination. Dyslipidemia refers to levels of one or more kinds of lipids in the blood (triglycerides > 150 mg/dL, cholesterol > 200 mg/dL, LDL > 115 mg/dL) and/or the use of statins and other lipid-lowering drugs. The diagnosis of diabetes was based on standard international criteria [9].For the HIV+ group, HIV viral load, CD4+ cell counts, total serum cholesterol, low-density lipoprotein (LDL) cholesterol, high-density lipoprotein (HDL) cholesterol, glycemia, triglycerides, and body mass index (BMI) were recorded at each visit.

Our study was approved by our ethics committee, Campania Nord, during the meeting on 12 October 2022. The project approval code is N. Reg. CECN 1930. All procedures performed in this study were in accordance with the 1964 Helsinki Declaration and its later amendments or comparable ethics standards. Informed consent was obtained from all participants included in the study.

### 2.1. Serological Analysis

The HIV viral load was assessed by real-time PCR with the lowest detection limit of 40 copies/mL. Lymphocyte subsets (CD4+, CD8+) were evaluated with flow cytofluorimetry using monoclonal antibodies and a fluorescence-activated cell sorter scan (Becton Dickinson, Mountain View, CA, USA). Serum lipids and routine analyses were performed, applying standard procedures.

### 2.2. Statistical Analysis

Continuous variables were described as means and standard deviation (SD) if normally distributed and as medians and interquartile range (IQR) if not normally distributed. To evaluate the distribution of continuous variables, we used the Shapiro–Wilk test, and to assess homoscedasticity, we used the Levene test. Categorical variables were described as absolute frequency (n) and percentage (%). The univariate tests used to compare HIV+ and HIV− groups were the Mann–Whitney test, the analysis of variance, and the Pearson chi-squared test (or Fisher test or Mantel–Haenszel test, as appropriate), respectively. The risk of having a pathological finding was evaluated by means of logistic regression (odds ratio, OR, and 95% confidence intervals, CI), univariate, and multivariate. In the multivariate equation, we included all variables significantly associated with the pathological findings in the univariate analysis or were different between groups. All *p*-levels were two-sided, at the significance level < 0.05. All statistical analyses were performed using SAS for Windows 9.4 (SAS Institute, Cary, NC, USA).

## 3. Results

The main characteristics of the two groups are summarized in Table 1. All patients were Caucasian. Patients in the HIV+ group were younger than the HIV− control group (49 years, IQR 38–59 vs. 70 years, IQR 63–75, *p* < 0.0001) and were more frequently male (79% vs. 57%, *p* < 0.0001). The HIV+ group showed a lower prevalence of hypertension (24.5% vs. 33.4%, *p* = 0.003) and diabetes (5.5% vs. 13.7%, *p* < 0.0001), and a higher frequency of dyslipidemia (55.8% vs. 31.5%, *p* < 0.0001), as compared to the HIV− group. A difference was seen in the frequency of lipid-lowering drug use since all HIV− people with dyslipidemia resulted in treatment, whereas the corresponding figure was a meager 25% for PLWH (*p* < 0.0001). Regarding antiretroviral regimens in the HIV+ group, patients were treated mainly with integrase strand transfer inhibitors (INSTI) (44.5%), with protease inhibitors (PI) in 17.4%, with non-nucleoside reverse transcriptase inhibitors (NNRTI) in 20.5%, dual regimens in 7.6%, and other regimens, treatment suspension, or regimen unknown, in 23.2% (Figure 2). The median treatment duration was 7.7 years. The median CD4+ count was 613 cells/μL, while a low CD4+ nadir (<100 cells/μL) was reported in 22.9% of patients (Table 1). Regarding the Doppler scan data of the supra-aortic vessels, the overall prevalence of plaques was lower in the HIV+ than in the HIV− group (33.4% vs. 43.9%, *p* < 0.0001). However, in strata of age, it became evident that the opposite was true, with the prevalence of plaques presence being consistently higher in HIV+ than in HIV− people (Figure 3 and Table 2). In Table 3, in the univariate analysis, we found, as expected, that younger age was protective for plaques presence, as well as being female (although not significantly), whereas hypertension (both treated and untreated), diabetes, and dyslipidemia were associated with a higher risk of plaques. Being HIV+ was, in the univariate analysis, associated with a lower risk of plaques (crude OR 0.64, 95% CI 0.49–0.84). Including the potential confounders in the logistic regression equation, we confirmed the protective role of younger age and that hypertension and dyslipidemia were associated with an increased risk of plaques. Sex and diabetes maintained similar estimated ORs but were not statistically significant anymore. On the contrary, the HIV+ status, after including the age in the model, reversed its estimated association and showed an increased risk of plaques (adjusted OR 1.61, 95% CI 1.06–2.44) (Table 3). Including lipid-lowering treatment in the model did not change the OR estimates in a significant way. Then, we repeated this analysis, splitting the HIV+ group by ART treatment duration into less than 5 years (*n* = 144, 37.9%), 5 to 10 years (*n* = 78, 20.5%), and more than 10 years (*n* = 158, 41.6%). The univariate analysis showed that HIV+ people with lower exposure to ART had a seemingly lower risk of plaques, whereas those with more than 10 years of ART exposure were similar to HIV− people in terms of risk. However, when including age and other confounders in this analysis, we found that increasing ART exposure in the HIV+ group was associated with an increasing risk of plaques, reaching statistical significance in the most exposed group (ART duration > 10 years, adjusted OR 1.93, 95% CI 1.21–3.10, *p* = 0.02) (Figure 4).

## 4. Discussion

Our study seems to suggest that HIV+ patients treated with antiretroviral regimens are at greater risk of developing atheromatous plaques compared to HIV-negative subjects. This risk appears higher for longer exposure to ART as an effect of cumulative metabolic toxicity on the onset of IMT and carotid plaques. In this regard, about the evidence of IMT and plaques in HIV patients, there is much data in the literature; however, they are often contrasting in their conclusions. In an older study from 2006, Jerico et al. evaluated the association between ART and subclinical carotid atherosclerosis in HIV-positive patients [10]. They enrolled 132 patients, 93 on ART and 39 never treated. They concluded that ART should be considered a strong independent predictor for the development of subclinical atherosclerosis in HIV-positive patients. In 2012, Spanish authors tried to explain in which way ART could determine IMT in HIV+ patients [11]. In their study, the authors described the association of HIV infection and antiretroviral therapy with levels of endothelial progenitor cells and subclinical atherosclerosis. They concluded that ART exposure is the main predictor of circulating vascular progenitor cell levels. Courier et al. analyzed the progression of cIMT in HIV-positive and HIV-negative adults [12]. They enrolled 134 patients, divided into 3 groups: a group composed of HIV-experienced patients treated with a protease inhibitor (PI) in their current regimen, a second group based on HIV-experienced patients treated without PI, and a third group of HIV patients. They concluded that there was no difference in the rate of progression of cIMT between PI-treated patients and PI-untreated patients and between HIV-positive cases and HIV-negative controls. The authors themselves attempted to explain these results, stating that they needed a larger sample to detect a clinically significant difference within the described range. Furthermore, the PI drugs used had a lower impact on lipid parameters and CVR than those adopted in other studies. Only 30% of subjects treated with PI were receiving a ritonavir-containing regimen at baseline. In our recent study, we evaluated the lipid profile and cIMT in antiretroviral-experienced HIV-positive patients treated with PI-based versus PI-sparing regimens [13]. Our real-life data show that patients treated with PI have a tendency to develop both increased dyslipidemia and increased pathological IMT and atheromatous plaques. These findings could be useful for optimizing antiretrovirals for patients with cardiovascular risk factors. In a research letter, French authors analyzed the change in the progression of atherosclerosis in HIV+ patients [14]. They enrolled 233 patients from the Aquitaine cohort, followed for 36 months. They showed that median IMT increased in the first 12 months and then decreased by month 36. Another French study evaluated the progression of cIMT over time in HIV-positive patients [15]. They enrolled 346 HIV+ patients and performed a Doppler scan of the supra-aortic trunks at baseline and after 12 months. They observed a significant but moderate increase in the median IMT of the common carotid artery (CCA), from 0.54 to 0.56 mm. At the same time, a significant association was found between cross-sectional CCA IMT measures at the 12-month control and conventional cardiovascular risk factors. A German study evaluated the relationship between HIV infection, antiretroviral drugs, and ultrasound evidence of early atherosclerosis in the context of vascular risk factors [16]. HIV infection and HAART are independent risk factors for early carotid atherosclerosis. The observed increase in IMT suggests that vascular risk is 4–14% greater and that “vascular age” is 4–5 years greater in HIV-positive individuals. A 2016 Brazilian study, in contrast to most data in the literature, showed that HIV infection is not associated with carotid intima-media thickness in Brazil [17]. They compared patients from 3 groups, one of 535 HIV+, 88 HIV-negative, and 10,943 participants in the ELSA-Brazil study. The authors show that cIMT in this context is not different in HIV-positive patients in Rio de Janeiro compared to two different groups of HIV-negative individuals. In a more recent study from 2018, the authors compared 428 HIV+ patients in a Swiss cohort and 276 HIV-negative controls to evaluate subclinical coronary disease [18]. Their data showed that HIV-positive people in Switzerland had a similar degree of non-calcified/mixed plaque and high-risk plaque and may have less calcified coronary plaque and lower coronary atherosclerosis involvement and severity score than HIV-negative people with similar Framingham risk scores. In contrast to this study, in a recent Canadian work, the authors evaluated 102 PLWH and 84 controls who performed ultrasound scans of the left and right common carotid arteries and internal carotid arteries [19]. The authors concluded that HIV status has been identified as a cofactor associated with carotid artery plaques. In a recent meta-analysis, Soares et al. demonstrated that people living with HIV have a higher prevalence of non-calcified coronary plaques and a similar prevalence of coronary artery calcium compared to HIV-negative individuals [20]. Our retrospective cross-sectional study was conducted to evaluate the role of HIV infection and antiretroviral therapy in determining a higher risk of IMT and atheromatous plaques. To analyze this aspect, we compared HIV-positive patients (cases) with HIV-negative patients (controls). The group of cases was enrolled from the Archiprevaleat cohort, based on HIV+ patients treated with ART, examined with Doppler scanning of the supra-aortic trunks. The control group was enrolled in a Vascular Surgery Unit of the University of Bari, based on patients without HIV infection but subjected to screening for IMT and plaques. At first glance, IMT and plaques were more evident in HIV-negative patients, who were, however, older than HIV+ patients. Aging, in this sense, is closely related to IMT and plaques. When the data were adjusted for age in the statistical analysis, the results changed, showing a clear correlation between HIV infection and the presence of IMT and plaques in the supra-aortic trunks. In logistic regression analysis, the risk of IMT and plaques was also related to the duration of antiretroviral treatment. Therefore, patients treated for more years were at greater risk of vascular lesions, a trend that increases with the duration of therapy and becomes significant for >10 years of ART. This aspect could be linked to the cumulative toxicity of the drug on the onset of vascular lesions. These data suggest that HIV and its treatment are statistically risk factors for IMT and plaques. These lesions are, at the same time, related to a higher cardiovascular risk. The onset of IMT and plaques is closely related to HIV infection and ART but also to the age of the patients and the duration of treatment. Consistent with our data, in a recent study, Majonga et al. analyzed cIMT in HIV-positive older children and adolescents on antiretroviral therapy [21]. A total of 117 participants with HIV and 75 healthy HIV-negative controls were included. Children with HIV taking ART have similar cIMT to HIV-negative children. These data seem to support the hypothesis that HIV and ART can induce IMT, especially if associated with a longer treatment history, as an effect of cumulative metabolic toxicity on the vessels, which is not evident in younger patients.

Our data confirm that myo-intimal media damages among HIV+ patients (both pathological thickness and plaques) could be due to the interplay of two major factors: the direct inflammatory effect of HIV infection and the effects of antiretrovirals, especially the older ones, like PIs and abacavir, characterized by higher cardiovascular toxicity. This makes HIV-positive patients, especially those previously submitted to treatment with older drugs, at higher CVR than the HIV-negative population.

There are some limitations in our study. Data could be more complete by enrolling a higher number of patients in the HIV+ group. At the same time, the control group belonged to a vascular clinic and had, therefore, a potentially higher incidence of CVR, likely having clinical indications to undergo a Doppler exam. In fact, people in the HIV-negative group were older than in the HIV+ one; moreover, traditional CVD risk factors, such as hypertension and diabetes, were more frequent in the control group. A CV risk factor more present in the HIV+ group was dyslipidemia, which was also more frequently untreated. This is likely due to the younger age of people living with HIV and to the pill burden already present in their treatment. All these differences, except the last one, should lead to an underestimate of the plaque risk in the HIV+ group; on the contrary, after accounting for all these variables, the risk of plaques and related CVR were higher in the HIV+ population than in HIV-negative controls, effectively underlining the role of the HIV infection and its treatment in the genesis of IMT.

## 5. Conclusions

In conclusion, we showed in a retrospective, cross-sectional study conducted on a large population that patients with HIV have a higher risk of IMT and plaques compared to HIV-negative patients. This appears to be equally due to HIV infection and its treatment and is more evident for patients treated for >10 years as an effect of cumulative metabolic toxicity on the onset of IMT and carotid plaques. Therefore, it is possible to conclude that HIV+ patients seem to have higher CVR than the HIV− population. This aspect is important because it could suggest that clinicians improve the prevention of heart disease in HIV+ patients who have a higher CVR. With this in mind, it would be appropriate to use the Framingham risk score not only to optimize antiretroviral regimens but also to perform a Doppler scan of the supra-aortic trunks in HIV+ patients.

## Figures and Tables

**Figure 1 biomedicines-12-00773-f001:**
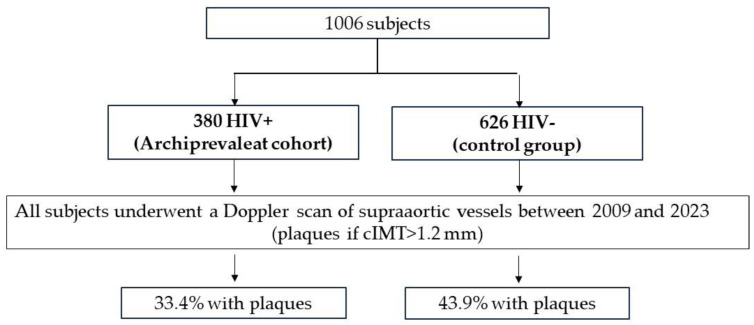
Study design.

**Figure 2 biomedicines-12-00773-f002:**
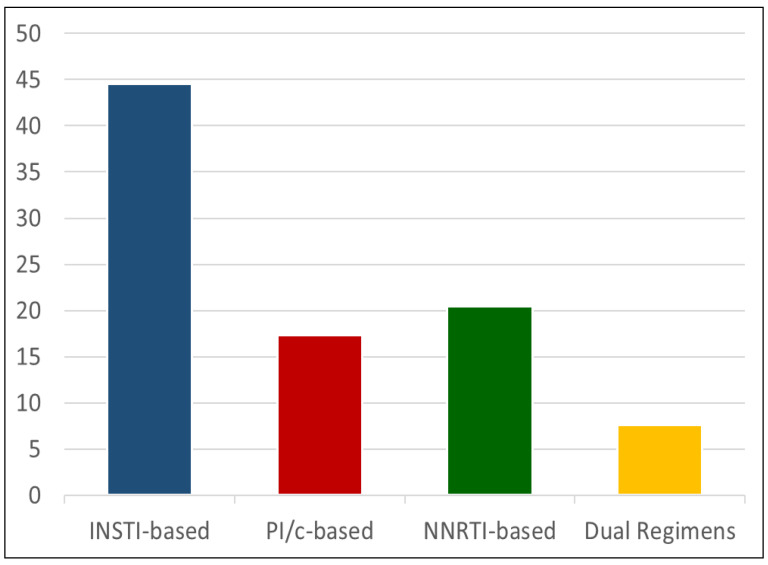
Antiretroviral regimens in HIV-positive patients. INSTI: integrase strand transfer inhibitors; PI: protease inhibitors; NNRTI: non-nucleoside reverse transcriptase inhibitors; dual: 2-drugs therapy.

**Figure 3 biomedicines-12-00773-f003:**
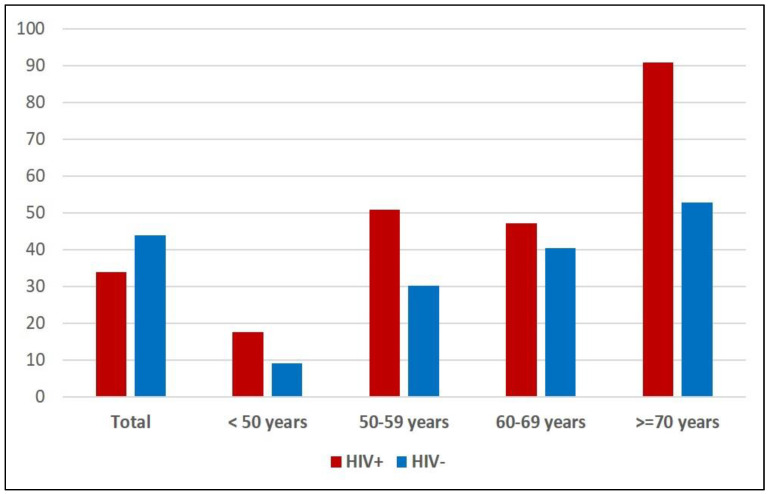
Prevalence of plaques in HIV-positive (*n* = 380) and HIV-negative (*n* = 626) people, total and by age class.

**Figure 4 biomedicines-12-00773-f004:**
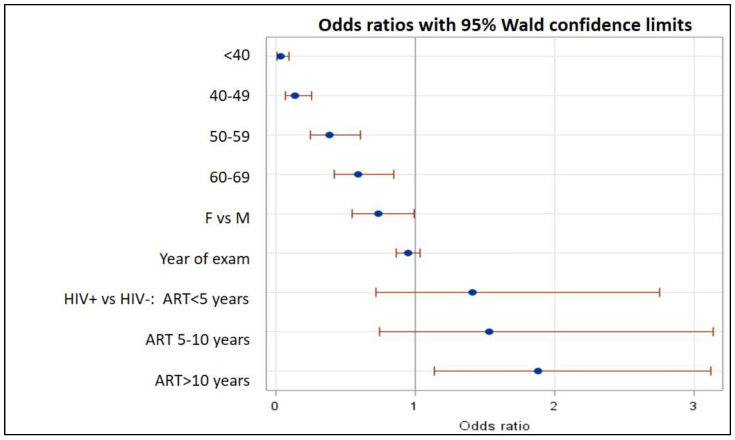
Odds ratios for risk of plaque (IMT > 1.2 mm) in HIV-positive patients compared to HIV-negative subjects.

**Table 1 biomedicines-12-00773-t001:** Characteristics of 380 HIV-positive and 626 HIV-negative patients undergoing intima-media thickness evaluation.

	HIV-Positive N = 380	HIV-Negative N = 626	*p*
Age, years, median (IQR)	49.0 (38.5–59.0)	70.0 (63.0–75.0)	<0.0001
Age class, years, n (%)			
<50	202 (53.2%)	31 (5.0%)	
50–59	118 (31.0%)	83 (13.3%)	
60–69	49 (12.9%)	183 (29.2%)	
≥70	11 (2.9%)	329 (52.6%)	<0.0001
Male, n (%)	300 (79.0%)	359 (57.4%)	<0.0001
Caucasian	380 (100%)	626 (100%)	1.0
Hypertension, n (%)	93 (24.5%)	209 (33.4%)	0.003
Antihypertensive treatment, n (%)	64 (16.8%)	192 (30.7%)	<0.0001
Untreated hypertension (only hypertensive), n (%)	29 (31.2%)	17 (8.1%)	<0.0001
Diabetes, n (%)	21 (5.5%)	86 (13.7%)	<0.0001
Dyslipidemia, n (%)	212 (55.8%)	197 (31.5%)	<0.0001
On lipid-lowering treatment, n (%)	52 (13.7%)	197 (31.5%)	<0.0001
Current NRTI, n (%)	247 (65.0%)	-	-
Current PI, n (%)	66 (17.4%)	-	-
Current NNRTI, n (%)	65 (17.1%)	-	-
Current INSTI, n (%)	168 (44.2%)	-	-
Years of ART, median (IQR)	7.7 (3.1–20.6)	-	-
CD4+, cell/µL, median (IQR)	613 (377–833)	-	-
Nadir CD4+ < 100 cell/µL, n (%)	87 (22.9%)	-	-

ART: antiretroviral treatment; IQR: interquartile range; INSTI: integrase strand transfer inhibitor; NNRTI: non-nucleoside reverse transcriptase inhibitor; NRTI: nucleoside reverse transcriptase inhibitor; PI: protease inhibitor.

**Table 2 biomedicines-12-00773-t002:** Doppler TSA data according to HIV status.

	HIV-Positive N = 380	HIV-Negative N = 626	*p*
Left IMT *w*/*o* plaques, mean (SD)	0.86 ± 0.17	0.93 ± 0.14	<0.0001
Right IMT *w*/*o* plaques, mean (SD)	0.83 ± 0.17	0.93 ± 0.14	<0.0001
People with plaques, n (%)	127 (33.4%)	275 (43.9%)	<0.0001
Left plaques, n (%) *	97 (25.5%)	215 (34.4%)	0.003
Right plaques, n (%) *	94 (24.7%)	231 (37.4%)	<0.0001
Left and right IMT ≤ 1.0 mm, n (%)	216 (56.8%)	276 (44.1%)	
Left and/or right IMT > 1.0 and ≤1.2 mm, n (%)	37 (9.7%)	75 (12.0%)	
Left or right IMT > 1.2 mm, n (%)	127 (33.4%)	275 (43.9%)	0.0001
People with plaques by age class, n (%)			
<50 years	34 (16.8%)	2 (6.4%)	0.14
50–59 years	60 (50.8%)	25 (30.1%)	0.003
60–69 years	23 (46.9%)	74 (40.4%)	0.41
≥70 years	10 (90.9%)	174 (52.9%)	0.01

IMT: intima-media thickness; SD: standard deviation. * the sum of patients with left and right plaques is higher than the number of patients with total plaques because of the concurrent presence of left and right plaques.

**Table 3 biomedicines-12-00773-t003:** Univariate and multivariate analysis for the risk of pathological IMT findings in HIV-positive as compared to HIV-negative people.

	Univariate Analysis			Multivariable Analysis		
	OR	95% CI	*p* Value	aOR	95% Confidence Interval	*p* Value
Age class, years, ref. ≥ 70						
<50	0.16	0.10–0.24		0.13	0.07–0.22	
50–59	0.62	0.44–0.88		0.49	0.31–0.75	
60–69	0.61	0.44–0.86	<0.0001	0.70	0.49–1.02	<0.0001
Female sex, ref. M	0.87	0.66–1.13	0.30	0.83	0.62–1.12	0.22
Hypertension, ref. No	2.95	2.24–3.90	<0.0001	2.08	1.54–2.80	<0.0001
Treated hypertension, ref. no hypertension	3.14	2.34–4.22		2.14	1.56–2.93	
Untreated hypertension, ref. no hypertension	2.11	1.16–3.85	<0.0001	1.79	0.95–3.38	<0.0001
Diabetes, ref. No	1.55	1.04–2.31	0.03	1.28	0.83–1.98	0.26
Dyslipidemia, ref. No	1.65	1.28–2.14	0.0001	1.57	1.17–2.10	0.002
HIV-positive, ref. HIV-negative	0.64	0.49–0.84	0.001	1.61	1.06–2.44	0.02
HIV-positive, ref. HIV-negative						
<5 years of ART	0.36	0.24–0.56		1.25	0.71–2.22	
5–10 years of ART	0.50	0.30–0.84		1.29	0.68–2.44	
>10 years of ART	1.10	0.77–1.56	<0.0001	1.93	1.21–3.10	0.05

aOR: adjusted odds ratio; ART: antiretroviral treatment; CI: confidence interval; OR: odds ratio. ref.: reference.

## Data Availability

The data presented in this study are available on request from the corresponding author.

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
