# Peer review of "Evaluation of Myo-Intimal Media Thickness and Atheromatous Plaques in People Living with HIV from the Archiprevaleat Cohort vs. HIV-Negative Subjects"

_biomedicines, 2024, doi:10.3390/biomedicines12040773_

Round 1
Reviewer 1 Report
Comments and Suggestions for Authors
The Authors aimed to evaluate whether HIV positive (HIV+) patients show high myo-intimal media thickness (IMT) and atheromatous plaques compared to HIV negative (HIV-) patients.
The article is written clearly. The results justify the conclusions.
Some minor remarks are as follows:
- In the Methods section the reference should be only marked with number, but fully cited in the Reference section: „The diagnosis of diabetes was based on standard international criteria (American Diabetes Association: Classification and diagnosis of diabetes. Diabetes Care 2016;39 (Suppl 1):S13–S22)”.
- A “p” level of significance is missing in the Statistical Analysis section.
- The limitations of the study need to be removed from the Conclusion section to the end of the Discussion section.
- The legend below Figure 2 needs to explain abbreviations.
- The abbreviations need to be explained when first mentioned in the text and should be used consistently thereafter.
- Minor editing of English language is recommended.
Comments on the Quality of English Language
Minor editing of English language required.
Author Response
The Authors aimed to evaluate whether HIV positive (HIV+) patients show high myo-intimal media thickness (IMT) and atheromatous plaques compared to HIV negative (HIV-) patients.
The article is written clearly. The results justify the conclusions.
- Thank you for your kind evaluation.
Some minor remarks are as follows:
In the Methods section the reference should be only marked with number, but fully cited in the Reference section: „The diagnosis of diabetes was based on standard international criteria (American Diabetes Association: Classification and diagnosis of diabetes. Diabetes Care 2016;39 (Suppl 1):S13–S22)”.
- Thank you for this remark, we corrected this reference.
A “p” level of significance is missing in the Statistical Analysis section.
- We added the P level: “All p-levels were two-sided, at the significance level <0.05. All statistical analyses were performed using SAS for Windows 9.4 (SAS Institute, Cary, NC).”
The limitations of the study need to be removed from the Conclusion section to the end of the Discussion section.
- Thank you, we inserted the limitations where appropriate.
The legend below Figure 2 needs to explain abbreviations.
- Thank you, we added the explanations and also modified the Figure to give more precise information.
The abbreviations need to be explained when first mentioned in the text and should be used consistently thereafter.
- Thank you, we checked the text.
Minor editing of English language is recommended.
- The text has been revised. We hope the English language is now adequate.
Reviewer 2 Report
Comments and Suggestions for Authors
First, a technical note - the percentage of article similarity = 35%, a relatively high result. IMT, serological analysis - these sections should be prescribed. The abbreviation "PLWH" is not known to all readers, so I suggest changing the article's title in this context. Since "the study aimed to assess the incidence of plaques in this cohort of PLWH compared to a group of HIV-infected people, to examine the potential role of HIV infection and ART treatment, beyond the classic factors of CVR - it is important that the study groups were relatively similar to each other. I am not sure whether the purpose of the study was formulated correctly. How did the authors examine the potential impact of HIV infection and ART treatment on the incidence of dental plaque? If it was just one assessment for HIV+ patients - the goal was not achievable. Since there were no studies before ART, I have doubts about the basis on which researchers assume that the drug's effect is crucial and influences the obtained result, and not, for example, individual characteristics and the patient's aging. Perhaps I missed something, but it would have been helpful to describe the study's limitations in more detail. I'm not sure if the group was recruited correctly. As is known, age itself affects IMT, and in this study, there is a very large, statistically significant difference between the age of HIV-infected patients and the HIV-infected control group. Comparison is difficult in this situation and may affect the conclusions. I am particularly concerned about the situation of the group of people aged 70 and over; in the case of HIV+ it is only 2.9% and in the case of the control group 52.6%. Wouldn't it be better to reduce the control group and select it appropriately using statistical methods to eliminate the influence of age on the obtained result? A similar doubt arises regarding statistically significant differences due to gender (p<0.0001). The difference, although much smaller, is the occurrence of diabetes and lipid disorders - which, as we know, also have a key impact on IMT levels. Dig. 4 - what do the authors mean when they write "increased risk of plaques?" What criterion did they use to create the board (IMT>...mm)? Moreover, this chart does not consider gender differences and compares, for example, 11 people and 329 people over 70 years of age, i.e., 202 vs. 31 in the age group <50 years or 60-69 years (49 people vs. 183). The size of the study groups should be considered in this figure. The conclusions do not convince me because they seem very pious. Many phrases raise some doubts: "at a higher risk of developing IMT." IMT measures the thickness of the intima-media complex - so you cannot write "developing IMT."; e.t.c... Moreover, what about the treatment of lipid disorders in the study groups? I have another comment regarding the HIV-tested group. These patients were referred for Doppler examination, meaning they had certain tendencies to abnormalities. And as I wrote several times above, they were definitely older than the HIV+ group. This means it is very difficult to comment on the results the researchers presented clearly. These comparison groups do not make good comparison groups.
My last question concerns the mechanism, and how the authors would like to explain the impact of drugs and HIV on the progression of atherosclerotic plaque.
Comments on the Quality of English Language
Moderate editing of the English language is required.
Author Response
First, a technical note - the percentage of article similarity = 35%, a relatively high result. IMT, serological analysis - these sections should be prescribed.
- We modified these sections, shortening and adding the reference to a previous paper with the complete description of our methods.
The abbreviation "PLWH" is not known to all readers, so I suggest changing the article's title in this context.
- We modified the title accordingly: “Evaluation of myo-intimal media thickness and atheromatous plaques in people living with HIV from the Archiprevaleat cohort vs HIV-negative subjects.”
Since "the study aimed to assess the incidence of plaques in this cohort of PLWH compared to a group of HIV- infected people, to examine the potential role of HIV infection and ART treatment, beyond the classic factors of CVR” - it is important that the study groups were relatively similar to each other. I am not sure whether the purpose of the study was formulated correctly.
- We modified the sentence adding “negative” to the control group: “The aim of the present study is to evaluate the prevalence of plaques in this cohort of PLWH, as compared to a group of HIV-negative subjects, investigating the potential role of HIV infection and ART treatment, besides that of the classic CVR factors”.
How did the authors examine the potential impact of HIV infection and ART treatment on the incidence of IMT plaque? If it was just one assessment for HIV+ patients - the goal was not achievable.
- We hope that is now clear that the control group was HIV-negative.
Since there were no studies before ART, I have doubts about the basis on which researchers assume that the drug's effect is crucial and influences the obtained result, and not, for example, individual characteristics and the patient's aging.
- Regarding the effect of drugs on IMT, if HIV+ patients on treatment have higher IMT values ​​than uninfected patients and if this is more evident for those treated for > 10 years, it is a clear suggestion that ART may play a role. Our data and our hypotheses are in line with several works in the literature cited in the text. The HIV+ status, after including the age in the logistic regression equation, showed an increased risk of plaques.
Perhaps I missed something, but it would have been helpful to describe the study's limitations in more detail. I'm not sure if the group was recruited correctly. As is known, age itself affects IMT, and in this study, there is a very large, statistically significant difference between the age of HIV-infected patients and the HIV-infected control group. Comparison is difficult in this situation and may affect the conclusions.
- Regrettably, the only comparison group without HIV infection was constituted by people who needed a IMT examination because of clinical symptoms or control due to old age. The population of PLWH, although ageing, is currently much younger than the control group. We used the appropriate statistical methods to account for this difference.
I am particularly concerned about the situation of the group of people aged 70 and over; in the case of HIV+ it is only 2.9% and in the case of the control group 52.6%. Wouldn't it be better to reduce the control group and select it appropriately using statistical methods to eliminate the influence of age on the obtained result?
- We could not match the subjects by age, because the overlap was limited to the central age class: we would have lost most people <50 and >70 years old. Despite this imbalance, the IMT was more evident in the HIV+ patient group and the difference with the HIV- control group, after correcting for age in the multivariate analysis, could even be underestimated.
A similar doubt arises regarding statistically significant differences due to gender (p<0.0001). The difference, although much smaller, is the occurrence of diabetes and lipid disorders - which, as we know, also have a key impact on IMT levels.
- The difference in the gender between HIV+ and HIV- population is related to the characteristics of PLWH population: overall, female sex represents between 25% and 30% of PLWH. So, it was expected to have less female subjects in HIV+ group than uninfected controls. As expected, the HIV-negative control group, being older, presents a higher prevalence of hypertension and diabetes, whereas dyslipidemia was more frequent in the HIV+ group, likely due to the effect of antiretrovirals. These factors had been also considered in the multivariate analysis.
Fig. 4 - what do the authors mean when they write "increased risk of plaques?" What criterion did they use to create the board (IMT>...mm)? Moreover, this chart does not consider gender differences and compares, for example, 11 people and 329 people over 70 years of age, i.e., 202 vs. 31 in the age group <50 years or 60-69 years (49 people vs. 183). The size of the study groups should be considered in this figure.
- Thank you for this remark. Indeed, we modified the caption in the current version: “Odds ratios for risk of plaque in HIV-infected patients compared to HIV-negative subjects.” The following sentences should have been deleted because comments that were included in the Results section. The IMT thickness for a plaque was defined in the text and in Table 2, however we also added the information in the Figure caption. As regards the size of different groups, this analysis compared all patients in the different age groups, thus, when evaluating the ORs for age, all patients (HIV+ and HIV-) are included in the age groups, and ORs are adjusted for their HIV status (and other significantly different characteristics), as well as females are compared to males, accounting for their age and HIV status (and other significantly different characteristics) and so on. The size of the groups, for each characteristic (age, sex, duration of treatement), is approximately related to the width of the confidence interval (the wider the 95% CI bar, the smaller the group).
The conclusions do not convince me because they seem very pious. Many phrases raise some doubts: "at a higher risk of developing IMT." IMT measures the thickness of the intima-media complex - so you cannot write "developing IMT."; e.t.c...
- Thank you, we deleted these words.
Moreover, what about the treatment of lipid disorders in the study groups?
- We added two sentences in the Results section to underscore this difference and commented in the Limitations section: “There are some limitations in our study. Data could be more complete by enrolling a higher number of patients in the HIV+ group. At the same time, the control group belonged to a vascular clinic and had therefore a potentially higher incidence of CVR, likely having clinical indication to undergo a Doppler exam. In fact, people in the HIV-negative group were older than in the HIV+ one; moreover, traditional CVD risk factors, such as hypertension and diabetes, were more frequent in the control group. A CV risk factor more present in PLWH was dyslipidemia, that is often associated to ART and is also more frequently untreated: this is likely due to the younger age of PLWH, and to the pill burden already present in their treatment. All these differences, except the last one, should lead to find lower incidence of plaque in PLWH; on the contrary, after accounting for all these variables, the risk of plaques and related CVR resulted higher in the HIV+ population than HIV uninfected controls.”
I have another comment regarding the HIV-tested group. These patients were referred for Doppler examination, meaning they had certain tendencies to abnormalities. And as I wrote several times above, they were definitely older than the HIV+ group. This means it is very difficult to comment on the results the researchers presented clearly. These comparison groups do not make good comparison groups.
- We added these limitations. However, as previously stated, although subjects in the control group were older and belonged to a vascular clinic, so with potential higher risk of IMT, data of our study show the contrary evaluating all variables in the multivariate analysis. So, it is possible to hypothesize that the risk of IMT for HIV+ patients is even underestimated. Thus, we can have a certain degree of confidence that HIV infection and ART conferred a supplementary risk for plaques.
My last question concerns the mechanism, and how the authors would like to explain the impact of drugs and HIV on the progression of atherosclerotic plaque.
- We added a paragraph with the rationale for the potential impact of ART drugs on the progression of atherosclerotic plaque: “Our data confirm that myo-intimal media damages among HIV+ patients (both pathological thickness and plaques) could be due to the interplay of two major factors: the direct inflammatory effect of HIV infection and the effects of antiretrovirals, especially the older ones, like PIs and abacavir, characterized by a higher cardiovascular toxicity. This makes HIV+ patients, especially those pre-viously submitted to treatment with older drugs, at higher CVR than HIV- population.” Furthermore, the role of HIV and ART in the genesis and progression of plaques and the potential CVR of this condition is clearly evident in several studies in the literature as reported by us in the bibliography. We added, anyway, in the text and in the references another study about the association of HIV-Infection and antiretroviral therapy with levels of endothelial progenitor cells and subclinical atherosclerosis [12].
Comments on the Quality of English Language
Moderate editing of the English language is required.
- The text has been revised. We hope the English language is now adequate.
Round 2
Reviewer 2 Report
Comments and Suggestions for Authors
The authors corrected all the shortcomings I pointed out. I have no further objections, and I believe that the current version of the article is suitable for publication.